# ON THE EVOLUTION OF NEURON COMMUNITIES IN A DEEP LEARNING ARCHITECTURE

## ABSTRACT

Deep learning techniques are increasingly being adopted for classification tasks over the past decade, yet explaining how deep learning architectures can achieve state-of-the-art performance is still an elusive goal. While all the training information is embedded deeply in a trained model, we do not yet understand much about its performance by only analyzing the model. This paper examines the neuron activation patterns of deep learning-based classification models and explores whether the models' performances can be explained through neurons' activation behavior. We propose two approaches: one that models neurons' activation behavior as a graph and examines whether the neurons form meaningful communities, and the other examines the predictability of neurons' behavior using entropy. Our experimental study reveals that both the community quality and entropy can provide new insights into the deep learning models' performances, thus paves a novel way of explaining deep learning models directly from the neurons' activation pattern.

## 1 INTRODUCTION

Deep learning allows the researchers to engineer better features and representation of data through representation learning (LeCun et al., 2015). Despite the widespread usage of deep learning methods, it is still considered a black box, and there is a lack of understanding of the working procedure of the models (Rai, 2020; Oh et al., 2019; Tzeng & Ma, 2005; Rudin, 2019). Researchers often rely on intuition and domain knowledge when designing deep learning architectures (Shahriari et al., 2015). They use the models' accuracy and loss to evaluate the performance and tune the hyperparameters to optimize the models (Géron, 2019; Gigante et al., 2019). However, to gain a deeper insight into the model, it is crucial to look beyond its accuracy and loss, i.e., we need to design additional evaluation metrics that can provide a level of confidence in the model predictions.

Exploring network architecture helps explain why some models perform better than others. For example, by comparing the weight optimization against model architecture, Gaier & Ha (2019) showed that a model's performance depends mostly on its architecture. Zhang et al. (2018) created an explanatory graph that can disentangle different part patterns from feature maps of the convolutional neural network (CNN). Visual analytics tools have also been used to understand the architectures better (Hohman et al., 2019; Gigante et al., 2019). However, such approaches sometimes require domain knowledge, and are also subject to human interpretation as they do not provide quantitative measures for the evaluation.

**Motivation.** In this paper, our main focus is on classification models. A deep learning model that performs a classification task is optimized to minimize the difference between the model's prediction and the actual value. The prediction is recorded as accurate if it is above a threshold, but this process does not fully leverage the information hidden inside the architecture. The neuron activation pattern that appears during the training contains rich information that, if understood well, can potentially be combined with model prediction to provide a level of confidence in the prediction value. It may also allow us to design more efficient models and additional evaluation metrics to help reduce the number of false negatives and false positives. If it is possible to relate the model's performance with the neuron activation pattern, we may understand why some models perform better than others by analyzing the pattern. This motivated us to explore different ways to examine neurons' activation pattern and establish their relation to the training and test accuracy.

**Contribution.** We study the neuron activation pattern using two approaches: one based on a graph-theoretic model, and the other is based on an information-theoretic model. Our proposed metrics show a correlation with the accuracy on both benchmark and real-life datasets in both cases.

For the graph-theoretic approach, we created a novel 'activation pattern graph' model and showed that the set of neurons that are frequently and highly activated for a class often forms a community (i.e., an induced subgraph that contains more edges within than outside of the subgraph) in the activation pattern graph. We observed that modularity, a widely used metric to measure community quality, is closely related to the model's training and test accuracy.

For the information-theoretic model, we propose a novel method that allows us to measure the predictability of the neurons' activation behavior leveraging entropy (Gray, 2011). Our experimental results show that the entropy of the neurons is relevant to the model's training and test accuracy.

## 2 RELATED WORK

**Neuron Activation Pattern.** Researchers have investigated both neuron activation patterns and loss function to explain deep learning models. Olah et al. (2017) studied what the neurons respond to and proposed that neurons work in a group. Li et al. (2017) showed a relationship between the performance of the deep learning model and the convexity of the loss function. Mahendran & Vedaldi (2015) studied the feature maps of the CNN models to understand the activation pattern by inverting the models and showed that there is indeed a relation between the performance of the model and the activation pattern. Kim et al. (2018) vectorized the activation values of hidden layers and created striped patterns to find relevance between pattern and decision. However, they experimented with a small dataset without directions for generalizability.

Bau et al. (2017) studied the relation between the activation pattern and semantic concepts. They optimized the hyperparameters of the deep learning models and quantitively analyzed the effect of changing different parameters. The relevance between activation patterns and semantic concepts was further studied by Fong & Vedaldi (2018). They created vector response of semantic concepts based on activation pattern and established that the neurons work in a group, and the same neuron can represent multiple concepts. We found a similar characteristic of the neuron activation pattern in our study, where there was an overlap of neurons representing multiple classes. However, with training, a class was represented by more unique neurons.

Another way of interpreting the deep learning models' learning is by selecting image patches that maximize the neuron activation (Zeiler & Fergus, 2014). Zhou et al. (2014) described scenes using the units from the feature maps and also proposed that individual units behave as object detectors. Simon et al. (2014) analyzed the deep learning model's feature maps and found a spatial relation between the activation centers and the semantic parts or bounding boxes of the ground truth images. Zhang et al. (2018) used hierarchical explanatory graphs across layers to propose a way of maximizing a deep learning model's performance. Bau et al. (2020) discovered that individual units of CNN can learn concepts from the images and some of them strongly influence the decision of the model. Although there have been several attempts to explain activation patterns, in this paper we take a very different approach to the explanation of the neural network models by combining the graph-theoretic and information-theoretic approaches, which allow us to relate various quantitative measures to the model's performance.

**Visualization and Interpretability.** Visualization is considered to be a promising approach for explaining complex systems, and as a result, there have been several attempts to explain the deep learning models using visualization tools (Hohman et al., 2018; Chung et al., 2016). A dataflow diagram is a simple way to visualize a deep learning model's architecture (Wongsuphasawat et al., 2017). However, dataflow diagrams do not tell much about the model's learning process.

Smilkov et al. (2017) proposed "Tensorflow Playground," where users can train a model by changing the hyperparameters and observe the training process with various plots and charts. Harley (2015), proposed an interactive visual inspection tool, where a model was trained on the MNIST dataset (Le-Cun et al., 2010), and for input data, the tool showed which feature maps were activated and also which part of the input data was activating the maps. ACTIVIS (Kahng et al., 2017) is an interactive visualization tool that, along with the dataflow diagram, provides a projection of the neuron activation

for different instances. This only allows users to explore the activation pattern for different samples, classes, and subsets rather than explaining the model's working procedure. CNNComparator (Zeng et al., 2017) is a tool to compare the performance metrics and the distribution of hyperparameters of two deep learning models.

A rich body of research examines ways of visualizing feature maps of CNN models (Dobrescu et al., 2019; Simonyan et al., 2013; Zeiler & Fergus, 2014). A common limitation of studies that rely on visualization is that the visual assessment is subjective and often hard to evaluate quantitatively.

**Entropy and Deep Learning.** Entropy is commonly used to quantify errors, e.g., the cross entropy is a widely used loss function (Gordon-Rodriguez et al., 2020). Many researchers proposed modifications to this loss function. Martinez & Stiefelhagen (2018) proposed Tamed Cross Entropy, which has the same convergence property as the cross entropy but is more robust against uniformly distributed label noise, Zhou et al. (2019) proposed Maximum Probability based Cross Entropy (MPCE) loss function, which uses a MPCE based gradient update algorithm and has less back-propagation error than cross entropy, and Gordon-Rodriguez et al. (2020) proposed to use log-likelihood of the continuous categorical distribution in the place of the cross entropy loss used in label smoothing and actor-mimic reinforcement learning.

Barbiero et al. (2021) proposed an entropy-based linear layer for concept-based deep learning models, which utilizes entropy to choose limited subset of input concepts, allowing it to provide concise explanations of its predictions. Huo et al. (2020) implemented a maximum entropy regularizer that encourages uniform weight distribution. Li et al. (2020) approximated the gradient of the cross entropy loss function, which are robust against noise and can avoid the vanishing gradient problem.

Although entropy measures has been used for model optimization, we leveraged it to analyze the activation pattern of the models and propose performance metric that correlates to the model accuracy.

## 3 TECHNICAL BACKGROUND

**Graph and Community Structure.** A *graph* consists of a set of elements (nodes) and a set of pairs of elements (edges), where the nodes represent objects and edges represent pairwise relationships. A *community* of a graph is defined as a subgraph that contains more edges within than the edges connecting them to the rest of the graph. Community detection algorithms often define a quality measure for the communities and then attempts to find a partition of the nodes that maximizes the quality measure (Fortunato & Castellano, 2007; Newman & Girvan, 2004). Later, we will define graphs with nodes as neurons and edges as their simultaneous activation. We will examine the communities of these graphs based on several quality measures as explained in the next section.

**Modularity.** Modularity is a widely used metric to assess the quality of a given set of non-overlapping communities (Newman, 2004; Duch & Arenas, 2005; Clauset et al., 2004). The idea of modularity is based on comparing the given graph $G$ with a random graph. Given two communities $C_i$ and $C_j$ in $G$, we use the notation $e(C_i, C_j)$ to denote the number of edges between these two communities. Let $a_{C_i}$ be the fraction of all the edges connecting the community $C_i$ to all other communities, i.e., $a_{C_i} = \sum_j e_{C_i C_j}$. For a random graph, the fraction of the resulting edges that connect nodes within the community $C_i$ is $a_{C_i}{}^2$. Hence the modularity (Newman & Girvan, 2004) can be formalized as $Q_{no-overlap} = \sum_i (e_{C_i C_i} - a_{C_i}{}^2)$. This is commonly expressed leveraging the adjacency matrix of the graph (Blondel et al., 2008). Let $A_{vw}$ be the adjacency matrix of $G$, and let $m$ be the number of edges in $G$. For a vertex $v$ (resp., $w$) in $G$, we denote its degree and community by $k_v$ and $C_v$ (resp., $k_w$ and $C_w$). Then

$$Q_{no-overlap} = \frac{1}{2m} \sum_{v,w} (A_{vw} - \frac{k_v k_w}{2m}) \delta(C_v, C_w), \text{ where } \delta(C_v, C_w) = \begin{cases} 1, \text{ if } C_v = C_w \\ 0, \text{ otherwise.} \end{cases} \quad (1)$$

In Eq. 1, $\frac{k_v k_w}{2m}$ represents the probability that two nodes $v$ and $w$ are connected in a random graph, and $A_{vw}$ is 1 or 0 depending on whether $v$ and $w$ are adjacent in $G$ or not. The term $\frac{1}{2m}$ is for normalizing the modularity value, and $\delta(C_v, C_w)$ regulates the algorithm only to consider the edges of a specific community. A larger modularity score indicates better quality for the communities.

The modularity above is defined only on a partition, i.e., when the communities are disjoint. Shen et al. (2009) proposed an extension of Eq. 1 for undirected, unweighted graphs with overlapping

communities, as follows:

$$Q_{unweighted,overlap} = \frac{1}{2m} \sum_{v,w} (A_{vw} - \frac{k_v k_w}{2m}) \frac{1}{O_v O_w}, \tag{2}$$

where $O_v$ and $O_w$ is the number of communities containing node $v$ and $w$, respectively. Thus the modularity becomes larger when the communities have less overlap. Chen et al. (2010) proposed a further extension for weighted, undirected graphs with overlapping communities:

$$Q_{weighted,overlap} = \frac{1}{2m} \sum_{c \in C} \sum_{v,w} (A_{vw} - \frac{k_v k_w}{2m}) \alpha_{cv} \alpha_{cw}, \tag{3}$$

where $\alpha_{cv}$ is the belonging coefficient (Nicosia et al., 2009) for a community $c$ and vertex $v$ is defined as $\alpha_{cv} = \frac{k_{cv}}{\sum_{c \in C} k_{cv}}$ and $k_{cv} = \sum_{p \in c} W_{vp}$, where $C$ is the set of community in graph $G$ and $c \in C$, $W_{vp}$ is the weight of the edge between node $v$ and $p$. In Eq. 3, if node $v$ belongs to only one community $c$, $\alpha_{cv}$ is equal to 1; if node $v$ does not belong to community $c$, $\alpha_{cv}$ is equal to 0, which ensures the consistency of Eq. 3 with Eq. 1. Compared to Eq. 1, Eq. 3 considers overlapping communities similar to Eq. 2, but unlike Eq. 1 it considers the edge weights too. The benefit of Eq. 3 over other equations is, a pair of nodes within a community with higher edge weight contributes more to the quality measure compared to the pairs with low edge weight.

**Entropy.** Entropy is a metric that measures the level of uncertainty in a system, and a rich body of research examines different ways of measuring entropy (Borowska, 2015). Shanon entropy is a widely used metric in information theory (Shannon, 2001), which calculates the average information available based on the probability of a variable's possible outcomes. Let $X$ be a discrete random variable with possible outcomes $x_1, x_2, ..., x_k$ which occurs with probabilities $P(x_1), P(x_2), ..., P(x_k)$, then the Shanon entropy, $H$ of the variable $X$ is defined as follows:

$$H = - \sum_{i=1}^{k} P(x_i) \log P(x_i). \tag{4}$$

A higher value of Shanon's entropy indicates a more uncertain outcome, which is difficult to predict.

## 4 HYPOTHESIS

We assume that if a neuron is frequently activated with a high activation value for a particular class, then the neuron is a representative of that class. Consider a graph with nodes as neurons and edges representing simultaneous activation of pairs of neurons. We refer to such a graph as an 'activation pattern graph', and describe the details in Section 5.1. Although one neuron can be representative for multiple classes, over the training, we expect the class representatives to have less overlap. Hence we also expect the activation pattern graph to evolve such that the neurons representing a class form a community. Furthermore, over the training, the neurons are expected to have more certainty in their activation pattern for various classes (rather than being random). Therefore, we expect the 'activation pattern entropy', a measure related to the predictability of a neuron behavior, as described later in Section 5.2, to decrease. In particular, we examine the following hypotheses.

**H1:** The neurons that are frequently activated together form a community in activation pattern graph.

**H2:** The modularity of the activation pattern graph is related to a deep learning model's performance.

**H3:** The entropy of the activation pattern is related to a deep learning model's performance.

## 5 METHODOLOGY

In this section, we describe the activation pattern graph and activation pattern entropy, which are at the core of our methodology and experimental design.

### 5.1 ACTIVATION PATTERN GRAPH

Assume that there are $C$ classes in the training dataset, and let $D_i$, where $1 \leq i \leq k$, be the subset corresponding to the $i$th class. Let $\ell$ be a fully connected layer in the neural network with $n$ neurons.

By the notation $v_i(q, \ell, d)$, we denote the activation value of a neuron $q \in \ell$ for a data $d$ that belongs to the $i$th class. Thus the average activation value of a neuron $q$ in layer $\ell$ for a class $i$ is as follows:

$$V_i(q, \ell) = \frac{\sum_{d \in D_i} v_i(q, \ell, d)}{|D_i|}. \tag{5}$$

We construct the *activation pattern graph* $G$ of a layer $\ell$ as follows. We first calculate each neuron's average activation value for a class. Next, for every class, we select $S$ neurons with the highest average activation value and take their union to create the vertex set of $G$. Let $p, q$ be a pair of neurons in $S$ and let $\delta_{p,q}$ be the number of data elements in $D_i$, where both $p$ and $q$ obtain activation values that are larger than their individual average activation values, i.e., $\delta_{p,q} = |\{d : d \in D_i, v_i(p, \ell, d) > V_i(p, \ell)$ and $v_i(q, \ell, d) > V_i(q, \ell)\}|$. Then, for class $i$, we create an adjacency matrix, $A_{i,\ell} = (a_{p,q})_{n \times n}$, where

$$a_{p,q} = \begin{cases} \frac{\delta_{p,q}}{|D_i|}, \text{ if } p \in S \text{ and } q \in S \\ 0, \text{ otherwise.} \end{cases} \tag{6}$$

The final adjacency matrix, $A_\ell$ of the activation pattern graph is computed as $A_\ell = \sum_{i=1}^{k} A_{i,\ell}$. Thus intuitively, an activation pattern graph maintains a set of neurons that are frequently activated, where a weighted edge between a pair of neurons represents the frequency of their simultaneous activation. We created a graph for every iteration and every fully connected layer (except the output layer).

## 5.2 ENTROPY OF ACTIVATION PATTERN

In a neural network with fully connected layers and ReLU (Rectified Linear Unit) as the activation function, the activation of each neuron is calculated as $ReLU(\mathbf{y}) = max(0, \mathbf{y})$, where $\mathbf{y}$ is the output of a layer. Let $W_\ell$ and $\mathbf{B}_\ell$ be the weight and bias of layer $\ell$ of a neural network. Then for an input $I$, the output of the layer $\ell$ is calculated as $\mathbf{y}_\ell = (W_\ell \times I) + \mathbf{B}_\ell$. At the beginning of training, the weights and biases are randomly initialized, and over the training, these values are updated for improved predictions. The neuron activation thus gets more and more influenced by the classes present in the data. Due to random initial weights, the activation of the neurons also becomes unpredictable. In such a case, the activation patterns' entropy is high, reflecting the system's randomness. As the training progresses, the activation pattern is biased by the data, and thus the entropy will decrease.

We now compute the *activation pattern entropy* of a particular neuron, which is based on the idea of measuring the predictability of its activation value. We create a $|D| \times N$ activation pattern matrix, $F$, where $|D|$ is the size of the training dataset, and $N$ is the total number of neurons in the architecture, except for the neurons in the last layer (output layer). Each entry $(i, j)$ of $F$ contains the activation value of the $j$th neuron subject to the $i$th element of the dataset $D$. We then compute a normalized matrix $F_{norm}$ by dividing each column by the column sum, i.e., $F_{norm}(i, j) = \frac{F(i,j)}{\sum_{i=1}^{|D|} F(i,j)}$.

To examine the predictability of the activation value for $j$th neuron, we categorized its normalized activation values using $R$ equal size bins. In other words, we create a histogram for the $j$th column values of $F_{norm}(i, j)$. The intuition is that if the neuron's activation is unpredictable, then the histogram will not have well-defined maxima or contain many local maxima. Otherwise, it will be activated for one or only a few classes and likely to produce a global peak. There is an exception, where a neuron may never be activated and will create a global maxima at the bin that contains the 0 value. Therefore, to create the histogram, we only consider the non-zero activation values. Let $B_k$ and $H_k$, where $1 \leq k \leq R$, be the $k$th bin and its number of elements. Let $h_i$ be the normalized value, i.e., $h_i = \frac{H_k}{\sum_{i=1}^{R} H_i}$. We use the vector $\mathbf{F}_v = [h_1, \ldots, h_R]$ to compute the activation pattern entropy $E_j$ for the $j$th neuron. $E_j = -\sum_{i=1}^{R} -h_i \cdot \log(h_i)$. The activation pattern entropy, $E$, of a deep learning model over all the neurons in all the fully connected layers is calculated as $E = \sum_{j=1}^{N} E_j$.

## 6 DATASET AND MODEL ARCHITECTURE

In our study, we used three benchmark datasets: MNIST (LeCun et al., 2010), Fashion MNIST (Xiao et al., 2017), CIFAR-10 (Krizhevsky et al., 2009) and one real-life dataset Plant village (Mohanty, 2018). We also created MNIST Mixed and Fashion MNIST Mixed by randomizing the labels of the respective datasets to examine the reliability of the proposed metrics.

The architecture for each dataset is illustrated in Table 1. The MNIST, Fashion MNIST, CIFAR-10 datasets each consists of 10 classes. There are 60,000 grayscale $28 \times 28$ images for training and 10,000 for testing in MNIST and Fashion MNIST, and 50,000 grayscale $32 \times 28$ images for training and 10,000 for testing in CIFAR-10.

The Plant village dataset consists of 28693 segmented color images of the healthy apple, blueberry and cherry, corn, grape, orange, peach and pepper, soybean, strawberry and squash, and tomato classes, and 6890 testing images. We transformed the color images to grayscale and reshaped the images to size $256 \times 256$.

Table 1: Details of the architecture of the deep learning models for different datasets. In the model we only used fully connected (FC) and dropout layers.

| Dataset | Image Size | Fully Connected Layers | Output Layer | Iterations | Training Accuracy (%) | Testing Accuracy (%) |
|---|---|---|---|---|---|---|
| MNIST | 28×28 | 512, 512 | 10 | 20 | 99.56 | 98.24 |
| MNIST Mixed | 28×28 | 512, 512 | 10 | 20 | 12.44 | 9.85 |
| Fashion MNIST | 28×28 | 512, 512 | 10 | 20 | 91.80 | 88.9 |
| Fashion MNIST Mixed | 28×28 | 512, 512 | 10 | 20 | 11.02 | 10.12 |
| CIFAR-10 | 32×32 | 1024, 1024, 1024, 512, 256, 128 | 10 | 40 | 46.24 | 43.43 |
| Plant Village | 256×256 | 8192, 2048, 1024, 512, 256, 128, 64 | 9 | 100 | 80.65 | 79.55 |

For all the models, apart from the input and output layers, we only used fully connected layers and dropout layer. For the fully connected layers, we used ReLU as the activation function, and Softmax activation for the output layer. For each model, we collected the weight values from 20 iterations at a uniform interval, where each iteration represents training the model with the dataset once. We denote the corresponding activation pattern graphs as $G(1), G(2), \ldots, G(20)$. Since different models needed a different number of training iterations, this interval length varies, e.g., the CIFAR-10 model collected every two iterations, whereas the Plant Village model every five iterations. See the supplementary materials for detailed model architecture.

## 7 Result and Discussion

In this section, we describe the experimental results. To create the activation pattern graphs, we took 50 neurons per class (i.e., we choose $|S| = 50$ in Eq. 6) with the largest average activation value. We repeated the experiments for the top 25 and top 100 neurons per class and achieved similar results.

### 7.1 Community Formation (H1)

Let $G(k)$ be a pattern activation graph of a fully connected layer at iteration $k$ and let $N_i$ be the set of neurons which are *representative* of the $i$th class, i.e., neurons that are frequently and highly activated for that class. The *unique* neurons of a pair of classes $i$ and $j$ are the neurons that belong to both $N_i$ and $N_j$, but in no other set $N_k$ where $k \notin \{i, j\}$. In other words, these neurons represents both classes $i$ and $j$, but no other class. If $i = j$, then we obtain the unique neurons of class $i$. We observed that in most cases, the number of unique neurons for a single class increases with training, whereas unique neurons for a pair of distinct classes decrease. This behavior indicates that the overlap between the communities is decreasing and the classes are forming a stronger community among themselves. Table 2 compares the confusion metrics obtained from $G(1)$ and $G(20)$ for the MNIST dataset, where each entry at the diagonal of a matrix represents the unique neurons for a single class. We observed similar behavior for all the datasets.

*Training Accuracy and Average Community Size:* We used Gephi (Bastian et al., 2009), a widely used graph visualization software, to compute force directed layouts (Jacomy et al., 2014) of the activation pattern graphs of different iterations. The communities were detected based on the Louvain method (Blondel et al., 2008). Although the visualization revealed community structures in these graphs, the change in the communities over the number of iterations was not readily visible from the graph layouts (see the supplementary material). Hence, we quantitatively examined how the number of nodes per community varies over the training. For each dataset, we used Gephi's modularity function with a resolution value of 0.5 to find the communities in $G(1), G(2), \ldots, G(20)$. Table 3

Table 2: Two confusion metrics computed from layer 1 for MNIST dataset. Each matrix represents the number of unique neurons for pairwise classes. Darker red color represents a higher value.

| | Iteration 1 | | | | | | | | | Iteration 20 | | | | | | | |
|---|---|---|---|---|---|---|---|---|---|---|---|---|---|---|---|---|---|
| 20 | 1 | 9 | 8 | 7 | 14 | 11 | 7 | 7 | 4 | 30 | 0 | 7 | 5 | 3 | 6 | 7 | 4 | 5 | 2 |
| 1 | 11 | 14 | 17 | 4 | 7 | 8 | 10 | 19 | 7 | 0 | 23 | 7 | 12 | 4 | 4 | 4 | 9 | 13 | 10 |
| 9 | 14 | 13 | 17 | 5 | 9 | 16 | 4 | 15 | 5 | 7 | 7 | 11 | 13 | 8 | 3 | 13 | 7 | 16 | 8 |
| 8 | 17 | 17 | 11 | 4 | 19 | 6 | 4 | 14 | 5 | 5 | 12 | 13 | 10 | 1 | 18 | 3 | 4 | 13 | 7 |
| 7 | 4 | 5 | 4 | 11 | 11 | 11 | 14 | 14 | 28 | 3 | 4 | 8 | 1 | 19 | 5 | 10 | 10 | 12 | 24 |
| 14 | 7 | 9 | 19 | 11 | 7 | 6 | 6 | 22 | 12 | 6 | 4 | 3 | 18 | 5 | 12 | 7 | 5 | 15 | 10 |
| 11 | 8 | 16 | 6 | 11 | 6 | 16 | 5 | 7 | 3 | 7 | 4 | 13 | 3 | 10 | 7 | 18 | 3 | 10 | 8 |
| 7 | 10 | 4 | 4 | 14 | 6 | 5 | 14 | 8 | 20 | 4 | 9 | 7 | 4 | 10 | 5 | 3 | 21 | 10 | 15 |
| 7 | 19 | 15 | 14 | 14 | 22 | 7 | 8 | 3 | 20 | 5 | 13 | 16 | 13 | 12 | 15 | 10 | 10 | 5 | 21 |
| 4 | 7 | 5 | 5 | 28 | 12 | 3 | 20 | 20 | 4 | 2 | 10 | 8 | 7 | 24 | 10 | 8 | 15 | 21 | 7 |

Table 3: Number of nodes in different communities of the activation pattern graph (detected by Gephi). For each dataset, the communities of $G(1)$ and $G(20)$ are shown along with the PCC between the average node per community and training accuracy. Darker red color represents higher value.

| | MNIST | | | | | | | | | | Fashion MNIST Mixed | | | | | | | | |
|---|---|---|---|---|---|---|---|---|---|---|---|---|---|---|---|---|---|---|---|
| Iteration | C1 | C2 | C3 | C4 | C5 | C6 | C7 | C8 | PCC | Iteration | C1 | C2 | C3 | C4 | C5 | C6 | C7 | C8 | PCC |
| 1 | 35 | 33 | 35 | 34 | 28 | 25 | 25 | 26 | | 1 | 9 | 7 | 6 | 6 | 5 | 5 | 4 | 3 | |
| 20 | 38 | 37 | 36 | 36 | 34 | 33 | 31 | 30 | 0.52 | 20 | 10 | 7 | 7 | 7 | 7 | 6 | 5 | 5 | 0.28 |

| | MNIST Mixed | | | | | | | | | | CIFAR-10 | | | | | | | | |
|---|---|---|---|---|---|---|---|---|---|---|---|---|---|---|---|---|---|---|---|
| Iteration | C1 | C2 | C3 | C4 | C5 | C6 | C7 | C8 | PCC | Iteration | C1 | C2 | C3 | C4 | C5 | C6 | C7 | C8 | PCC |
| 1 | 7 | 5 | 5 | 5 | 4 | 4 | 4 | 3 | | 1 | 12 | 12 | 11 | 9 | 6 | 6 | 6 | 4 | |
| 20 | 13 | 12 | 8 | 7 | 7 | 5 | 5 | 4 | -0.27 | 20 | 39 | 27 | 26 | 25 | 14 | 11 | 10 | 10 | 0.80 |

| | Fashion MNIST | | | | | | | | | | Plant Village | | | | | | | | |
|---|---|---|---|---|---|---|---|---|---|---|---|---|---|---|---|---|---|---|---|
| Iteration | C1 | C2 | C3 | C4 | C5 | C6 | C7 | C8 | PCC | Iteration | C1 | C2 | C3 | C4 | C5 | C6 | C7 | C8 | PCC |
| 1 | 35 | 16 | 16 | 14 | 11 | 10 | 8 | 7 | | 1 | 40 | 26 | 26 | 21 | 6 | 4 | 3 | 3 | |
| 20 | 32 | 27 | 25 | 21 | 15 | 15 | 12 | 11 | 0.71 | 20 | 62 | 62 | 46 | 35 | 34 | 12 | 5 | 4 | 0.82 |

shows the number of nodes in the eight largest communities $C_1, C_2, \ldots, C_8$ in layer 2 for different datasets and for two different iterations 1 and 20, where the distribution of community sizes appears to be less skewed over the training. We then perform the Pearson Correlation Coefficient (PCC) between the training accuracy and average node per community over the 10 iterations. For most of the datasets, we observed the average node per community to have a positive correlation with the training accuracy (Table 3), i.e., the PCC correlation was higher than 0. However, the weak (near-zero) correlation coefficient for MNIST Mixed and Fashion MNIST Mixed represents the randomness of the formation of the communities of the model. This indicates that, for models with good performance, the number of well defined communities increases with training and supports hypothesis **H1**.

## 7.2 Modularity and Accuracy (H2)

So far, we have observed that community detection can reveal meaningful clusters in the pattern activation graph, i.e., clusters are related to representative neuron sets for different classes. We now examine the other side, i.e., can the representative neuron sets for different classes be seen as well as the defined communities? To assess this, we compute different modularity metrics (Eq. 1– 3) for the activation pattern graphs $G(1), \ldots, G(20)$. We also used different community detection algorithms available in the python library NetworkX (Hagberg et al., 2008) and iGraph (Csardi & Nepusz, 2006). Note that these algorithms take an initial partition of the neurons and then change those partitions to optimize some quality measure. Therefore, they are not suitable in this context, yet, the Kernighan Lin Bisection (KLB) (Kernighan & Lin, 1970) could generate consistent results.

We computed the Spearman correlation coefficient (SCC) and Pearson correlation coefficient (PCC) to examine the potential relation between modularity and training accuracy (Table 4). A positive correlation value indicates that the communities' quality is positively correlated with training accuracy. We observed the same relation with the test accuracy. A large number of strong positive correlation coefficients in Table 4 supports hypothesis **H2**. A stronger correlation is observed in later layers for all the dataset. For most datasets, the unweighted overlap (Eq. 2), and weighted overlap metric (Eq. 3) captures the relation better, which is expected since the no-overlap metric neither considers weight nor the community overlap.

Table 4: Spearman and Pearson correlation coefficient between modularity and training accuracy for different layers. Darker blue represents higher values and darker red represents lower values. (See the supplementary material for the relation with test accuracy.)

| | MNIST | | | | MNIST Mixed | | | | Fashion MNIST | | | | Fashion MNIST Mixed | | | |
|---|---|---|---|---|---|---|---|---|---|---|---|---|---|---|---|---|
| | L1 | | L2 | | L1 | | L2 | | L1 | | L2 | | L1 | | L2 | |
| | PCC | SCC | PCC | SCC | PCC | SCC | PCC | SCC | PCC | SCC | PCC | SCC | PCC | SCC | PCC | SCC |
| KLB | 0.24 | 0.16 | 0.50 | 0.61 | -0.76 | -0.81 | -0.73 | -0.84 | -0.04 | -0.10 | 0.42 | 0.44 | -0.03 | -0.07 | -0.44 | -0.38 |
| No-overlap | 0.22 | 0.19 | 0.23 | 0.19 | -0.44 | -0.55 | 0.46 | 0.37 | 0.42 | 0.46 | 0.62 | 0.61 | 0.28 | 0.40 | 0.14 | 0.29 |
| Unweighted, Overlap | 0.53 | 0.37 | 0.78 | 0.68 | -0.70 | -0.73 | 0.36 | 0.37 | 0.71 | 0.67 | 0.77 | 0.72 | 0.43 | 0.51 | 0.22 | 0.35 |
| Weighted, Overlap | 0.36 | 0.33 | 0.77 | 0.57 | -0.63 | -0.69 | 0.48 | 0.47 | 0.56 | 0.46 | 0.78 | 0.79 | 0.43 | 0.51 | 0.26 | 0.38 |

| | CIFAR-10 | | | | | | | | | | | |
|---|---|---|---|---|---|---|---|---|---|---|---|---|
| | L1 | | L2 | | L3 | | L4 | | L5 | | L6 | |
| | PCC | SCC | PCC | SCC | PCC | SCC | PCC | SCC | PCC | SCC | PCC | SCC |
| KLB | 0.16 | 0.20 | 0.68 | 0.74 | 0.69 | 0.73 | -0.18 | -0.13 | -0.29 | -0.40 | -0.01 | 0.06 |
| No-overlap | -0.05 | -0.11 | 0.62 | 0.66 | 0.29 | 0.27 | 0.06 | -0.05 | 0.01 | 0.03 | -0.49 | -0.44 |
| Unweighted, Overlap | 0.34 | 0.44 | 0.87 | 0.86 | 0.73 | 0.82 | 0.82 | 0.79 | 0.66 | 0.77 | -0.37 | -0.29 |
| Weighted, Overlap | 0.42 | 0.55 | 0.78 | 0.68 | 0.71 | 0.78 | 0.71 | 0.70 | 0.71 | 0.79 | 0.02 | -0.03 |

| | Plant Village | | | | | | | | | | | | | |
|---|---|---|---|---|---|---|---|---|---|---|---|---|---|---|
| | L1 | | L2 | | L3 | | L4 | | L5 | | L6 | | L7 | |
| | PCC | SCC | PCC | SCC | PCC | SCC | PCC | SCC | PCC | SCC | PCC | SCC | PCC | SCC |
| KLB | 0.22 | 0.16 | 0.57 | 0.62 | 0.76 | 0.61 | 0.84 | 0.83 | -0.16 | 0.02 | -0.34 | -0.30 | -0.21 | -0.19 |
| No-overlap | 0.24 | 0.15 | 0.05 | 0.03 | 0.40 | 0.42 | -0.15 | -0.10 | 0.04 | 0.00 | 0.03 | -0.17 | 0.46 | 0.58 |
| Unweighted, Overlap | 0.14 | 0.09 | 0.49 | 0.47 | 0.83 | 0.80 | 0.91 | 0.85 | 0.63 | 0.65 | 0.46 | 0.47 | 0.74 | 0.82 |
| Weighted, Overlap | 0.24 | 0.26 | 0.59 | 0.52 | 0.84 | 0.87 | 0.88 | 0.78 | 0.71 | 0.78 | 0.66 | 0.69 | 0.62 | 0.72 |

Table 5: Spearman and Pearson correlation coefficient between the entropy and training accuracy. Darker blue represents higher values and darker red represents lower values.

| | MNIST | MNIST Mixed | Fashion MNIST | Fashion MNIST Mixed | CIFAR-10 | Plant Village |
|---|---|---|---|---|---|---|
| PCC | -0.87 | 0.22 | -0.95 | 0.57 | -0.95 | -0.94 |
| SCC | -0.94 | 0.51 | -0.99 | 0.67 | -0.97 | -1 |

## 7.3 ENTROPY AND ACCURACY (H3)

A higher entropy value represents more randomness in the activation behavior of a neuron. At the beginning of the training, due to random weights, the activation of the neuron is random. So, the entropy of the activation pattern over all the fully connected layers should be higher. As the training progresses, the entropy should decrease, representing a biased neuron activation behavior.

Table 5 shows a negative correlation between the entropy of activation pattern and training accuracy over model training for most of the datasets. We observed the same relation with the test accuracy. Figure 1 shows the change of normalized entropy with training and testing accuracy for different iterations for all the datasets, where a clear relation between the entropy and model performance can be observed. However, MNIST Mixed and Fashion MNIST Mixed has positive correlation values, indicating the absence of activation pattern with training due to it's random labels, which is consistent with hypothesis **H3**. Note that the evidence for hypothesis **H3** is supportive of **H1** and **H2**, where we examined the community behavior of the neurons and observed better community structure except for MNIST mixed and Fashion MNIST mixed.

We also examine entropy change for individual classes (Table 6), where to compute the entropy of a class, we take the same approach as in Section 5.2, but use only the training data corresponding to that class. In this setting, we obtain meaningful results (i.e., a negative correlation between the entropy and training accuracy) only for all well-trained models (MNIST, Fashion MNIST, and Plant

Table 6: Spearman and Pearson correlation coefficient between entropy and training accuracy for individual classes. Darker blue represents higher values, and darker red represents lower values.

| | MNIST | | MNIST Mixed | | Fashion MNIST | | Fashion MNIST Mixed | | CIFAR-10 | | Plant Village | |
|---|---|---|---|---|---|---|---|---|---|---|---|---|
| Class | PCC | SCC | PCC | SCC | PCC | SCC | PCC | SCC | PCC | SCC | PCC | SCC |
| 1 | -0.55 | -0.67 | -0.06 | -0.29 | 0.1 | 0.21 | -0.13 | -0.4 | 0.22 | 0.17 | -0.85 | -0.89 |
| 2 | -0.54 | -0.71 | 0.18 | -0.16 | -0.24 | -0.45 | 0.77 | -0.15 | 0.7 | 0.66 | -0.86 | -0.93 |
| 3 | -0.22 | -0.31 | 0.47 | -0.02 | 0.37 | 0.22 | 0.47 | 0.31 | 0.38 | -0.1 | -0.46 | -0.42 |
| 4 | -0.35 | -0.7 | 0.36 | 0.11 | -0.57 | -0.84 | 0.25 | -0.18 | 0.13 | 0.16 | -0.88 | -0.92 |
| 5 | -0.71 | -0.73 | 0.33 | -0.1 | -0.41 | -0.30 | 0.48 | 0.11 | -0.40 | -0.53 | -0.39 | -0.25 |
| 6 | -0.55 | -0.52 | -0.14 | -0.07 | -0.67 | -0.67 | 0.33 | -0.17 | -0.5 | -0.51 | -0.86 | -0.86 |
| 7 | 0 | 0.16 | 0.19 | 0.44 | -0.83 | -0.95 | 0.67 | 0.17 | 0.59 | -0.17 | -0.87 | -0.89 |
| 8 | -0.09 | -0.29 | 0.09 | 0.33 | -0.43 | -0.62 | 0.7 | 0.3 | 0.16 | 0.19 | -0.91 | -0.89 |
| 9 | -0.67 | -0.8 | 0.46 | 0.62 | -0.9 | -0.96 | 0.63 | 0.33 | 0.01 | 0.14 | -0.81 | -0.86 |
| 10 | -0.39 | -0.77 | 0.09 | 0.01 | -0.67 | -0.60 | 0.90 | 0.86 | -0.18 | -0.26 | - | - |

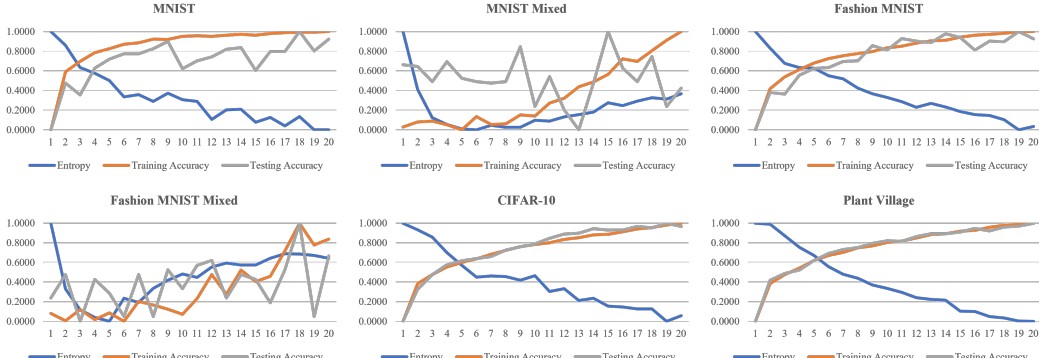

Figure 1: Change of normalized entropy, training and testing accuracy over iterations. Since values are normalized, a proper comparison should focus on trends instead of individual peaks or drops.

Village). However, there are few classes with positive and small negative correlation between entropy and accuracy. This is due to frequent change in the training accuracy of that class. For CIFAR-10, MNIST Mixed, Fashion MNIST Mixed, we observed a positive correlation values in most of the classes, and sometimes even found the correlation coefficient to be strongly positive. This is due to the inability of the model to separate the classes from one another.

## 8 LIMITATIONS AND FUTURE WORK

We have proposed novel methods to explain the behavior of neurons in deep learning models as the model training progresses. We used graph theoretic and entropy-based methods to model the activation pattern of the neurons. We quantitatively showed that neurons that are highly activated for a class form a community in our graph model. For the entropy-based approach, we analyzed the activation pattern using the Shanon entropy and found the entropy to show a negative correlation with the training accuracy.

There are many scope for future research. We used deep learning models with only fully connected layers; using different model architecture may help understand the entropy and modularity behavior with more granularity. Although we examine a diverse set of datasets, adding more datasets could strengthen the results. Our experimental results show that the modularity of the activation pattern graph and entropy of the activation pattern is related to the model's performance. Although we used widely used quality measures for the modularity and entropy, there is still scope for designing better quality metrics specifically for neural network context. Another exciting direction of study can be to use the modularity and entropy as a regularization in the loss function (Leavitt & Morcos, 2020) to better optimize the performance of the deep learning models. We believe that our results will inspire further research to explain the deep learning models using graph and information theoretic methods.

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
