# OpenReview forum: "On the Evolution of Neuron Communities in a Deep Learning Architecture"
_ICLR.cc/2022/Conference — ICLR 2022 Submitted_

### Official Review · Reviewer_RKXh · 2021-10-23

**Correctness:** 2
**Technical Novelty And Significance:** 1
**Empirical Novelty And Significance:** 1
**Recommendation:** 3
**Confidence:** 3

**Main Review:**

Understanding why deep learning works by inspecting the internal representations beyond loss value or performance (accuracy) is an important topic. This paper tries to advance our understanding with two newly proposed metrics, which is a good point. The writting and logic is clear.

However, several major concerns are obvious:

1. Not enough literature survey. The authors claim this paper "lay the initial groundwork for graph and entropy-based studies to analyze the deep learning models’ performance", but entropy is prevalent in deep learning. Given the bold literature on both understanding deep learning and entropy utilization, it is hard to say this paper is the first to do so.

2. No baseline methods to compare the significance of the two proposed metrics. There are many existing approaches in explaining deep learning models, such as turning deep models into decision trees in "Distilling a Neural Network Into a Soft Decision Tree". Then complexity measurement of the decision tree can be a baseline. Or counting the number of linear regions induced by neural networks like "On the number of linear regions of deep neural networks" is also a feasible idea. Frankly speaking, I'm not an expert in explaining deep learning, but I believe there have exist many methods to explain deep models' performance. The significance of methods proposed in this paper should be compared against existing approaches.

3. Experiments are rather preliminary, which only involves some toy datasets and MLP networks. It is not clear whether the method scales to larger and real-world problems, as admitted by authors in the paper.

By the way, I'm curious about how the experiments are done. In section 6, what do you mean by "iteration"? Is it one mini-batch or one epoch? What is the batch-size? The results are quite strange because I see that 99.56% accuracy is achieved within just 20 iterations.

**Summary Of The Paper:**

An important problem of understanding the performance of deep learning models is studied by this submission. Two graph-theoretic and information-theoretic metrics are checked in some datasets with MLP networks. Preliminary results with expected correlation are observed.

**Summary Of The Review:**

I acknowledge the importance of the research direction in this paper. The contribution of this paper is not properly compared against existing methods, making it difficult to tell their significance. Meanwhile, the toy setting in experiments makes me doubt whether the techniques can be extended to real-world scenarios.

---

> ### Author Response · Authors · 2021-11-16
> **Response to Reviewer RKXh's comments**
>
> We sincerely thank the reviewer for the insightful feedback and for appreciating our approach of proposing new performance matrices to advance the understanding of deep learning models.
>
> The reviewers pointed out other entropy based works in the literature. The entropy based performance metric is a novel contribution, which sets our work apart from previous researches. We appreciate the suggestion of the reviewer, and we should revise the paper to soften our claim on entropy and include some comments describing differences from prior research.
>
> We thank the reviewer for suggesting Distilling a Neural Network Into a Soft Decision Tree (Forsst and Hinton 2017) and On the number of linear regions of deep neural networks (Montufar et al. 2014) as a baseline for our work. However, these papers have different goals (e.g., Forsst and Hinton used decision trees to explain the decision of a deep learning model, and  Montufar et al. investigated the behavior of units in higher layers) than ours, which is specifically focused on designing performance metrics based on activation pattern. As accuracy is considered to be the most popular performance metric, we considered it to be the baseline and compared our metrics with it. However, we acknowledge that we can investigate these studies in our future work.
>
> The reviewer showed concern about whether the method scales to larger and real-world problems. Our activation pattern graph for current models is non-trivial as it carefully minimizes the impact of redundant edges to have the modularity signature revealed by the standard modularity measures. So, the performance of the proposed metrics should be consistent for larger models too. However, including larger datasets and more complex datasets will be a natural step to move forward.
>
> The reviewer wanted some clarification of the model's hyperparameters of our study. In each iteration, we trained the model with the whole dataset with a batch size of 32. We used Keras to build and train the models with the default learning rate. We also used Dropout layers in our models (see Figure 1 in Supplementary Document). To ensure the reproducibility of our experiments, we have shared our codebase in the supplementary materials. If necessary, we can share the saved models for the convenience of the reviewer.

---

### Official Review · Reviewer_Gvju · 2021-10-31

**Correctness:** 3
**Technical Novelty And Significance:** 3
**Empirical Novelty And Significance:** 3
**Recommendation:** 5
**Confidence:** 3

**Main Review:**

The authors propose two methods to analyze the behavior of neurons in neural nets. The main idea is to study the neuron activation
patterns of classification models and explore if the performance can be explained through neurons' activation behavior. The authors propose two approaches: one that models neurons' activation behavior as a graph and examines whether the neurons form meaningful communities, and the other that examines the predictability of neurons' behavior using entropy.

The paper is interesting, easy to read, and very well-structured. Furthermore, the subject of the paper is very relevant: the exploration of tools to increase the understanding and explainability of neural models. However, I have several important questions/comments to the authors:

1) In page 6, the authors clearly state that "we only used fully connected (FC) and dropout layers". However, in the supplementary material (page 1), the authors present a graphical representation of the deep learning model architecture used, and they include convolutional blocks. Three questions in this regard:
- do the authors employ or do not employ only fully connected layers?
- if what is stated in the paper is correct, why to employ only fully connected layers if the problem tackled is image classification? I think it would be more natural to employ ConvNets.
- do the authors think that the conclusions extracted would also apply to ConvNets and/or other neural models and problems?

2) The authors discuss three main hypotheses:
H1: The neurons that are frequently activated together form a community in the activation pattern graph.
H2: The modularity of the activation pattern graph is related to a deep learning model's performance.
H3: The entropy of the activation pattern is related to a deep learning model's performance.

Regarding H1, the authors conclude that "for models with good performance, the number of well defined communities increases with training". In turn, H2 can contribute to find representative neuron sets for different classes, and authors conclude that "the communities' quality is positively correlated with training accuracy". However, if I'm not mistaken, the main conclusion related to H3 is that, at the beginning of the training, due to the randomness of the initial weights, the activation of the neurons becomes unpredictable (high entropy); while, as the training moves forward, the activation pattern is biased and thus the entropy will decrease. It is not totally clear to me how this hypothesis (H3), and the associated experimentation and conclusions, effectively contribute to a better understanding and evaluation of the performance and explainability of neural networks.

3) Figure 3 of the supplementary material shows that the entropy, first, decreases with iterations (which fits the hypothesis managed in the paper: "As the training progresses, the entropy should decrease, representing a biased neuron activation behavior") but from the 8th-9th iteration it dramatically increases. What is the explanation of the authors for this, and how this fits the initial assumption?

4) All experimental evaluation is performed on quite "small" architectures. Is it possible that the explanatory and exploratory capacities of the presented methods are diminished with much larger/deeper models?

5) Also regarding the experimental configuration, it is unclear what exactly an "iteration" is, or what are the different hyperparameters involved (learning rate, batch size, etc. and how these were selected). In other words, I have the feeling that the results are hardly reproducible. On the other hand, there are no competitor approaches to compare the performance obtained by these metrics/methods. It would be interesting to see how the techniques presented in this paper compare to other techniques already present in the literature.



**Summary Of The Paper:**

The authors propose two methods to analyze the behavior of neurons in neural nets. The main idea is to study the neuron activation
patterns of classification models and explore if the performance can be explained through neurons' activation behavior. The authors propose two approaches: one that models neurons' activation behavior as a graph and examines whether the neurons form meaningful communities, and the other that examines the predictability of neurons' behavior using entropy.

**Summary Of The Review:**

Interesting paper and relevant subject. But I think there are some inconsistencies in the paper, and the overall impact, conclusions and applicability of the paper is somewhat dubious.

---

> ### Author Response · Authors · 2021-11-16
> **Response to Reviewer Gvju's comments**
>
> We sincerely thank the reviewer for the insightful feedback and for appreciating our approach to studying the classification models and explaining the performance of the model using modularity and entropy.
>
> The reviewer raised some questions as we mentioned fully connected layers as Convolutional Layer in Fig 1 of the supplementary material. This was a typing mistake from our end. We apologize for the inconvenience.
>
> The reviewer asked an excellent question of whether the technique can be used for ConvNet with image dataset. As ours is among the first line of investigation towards explaining neural networks leveraging modularity and entropy, we first looked at networks with only fully connected layers. Whether this could be generalized to CNN and if so then how the metrics would be redesigned is a natural future step for exploration.
>
> The reviewer mentioned that although H1 and H2 help understand the community structure, H3 does not directly produce any insights on neuron communities. The reason we examined H3 is that it provides us with an inherent idea of why the results for community structures (H1 and H2) should be valid. The activation pattern graph design is expected to have a more specialized structure when entropy negatively correlates with training accuracy. We should revise the discussion on H3 to reflect this.
>
> About the question on Figure 3 of the supplementary material, this is for a network trained with random labels (also see Table 4). Therefore, it is expected that the activation of the neurons will be random in the long run. Although the reviewer's observation is interesting, we think instead of focusing on specific iteration, the interpretation should be based on the trend.
>
> In response to the reviewer's concern on whether explanatory and exploratory capacities of the presented methods are diminished with much larger/deeper models, we think it will not be diminished. Because our activation pattern graph for current models are non-trivial as it carefully minimizes the impact of redundant edges to have the modularity signature revealed by the standard modularity measures. So, a deeper or wider model will not impact the findings.
>
> The reviewer wanted some clarification of the model's hyperparameters of our study. In each iteration, we trained the model with the whole dataset with a batch size of 32. We used Keras to build and train the models with the default learning rate. We also used Dropout layers in our models (see Figure 1 in Supplementary Document). To ensure the reproducibility of our experiments, we have shared our codebase in the supplementary materials. The reviewer excellently pointed that there are no competitor approaches to compare the performance obtained by the proposed metrics/methods, that's we compared the results with the accuracy of the model.

---

> > ### Comment · Reviewer_Gvju · 2021-11-30
> > **Response to author's feedback**
> >
> > I thank the authors very much for their detailed and elaborated response to my comments, questions and concerns. However, I must admit that my feeling that this paper, in its current state, is below the acceptance threshold persists. I encourage researchers to continue researching along these lines, and to try to incorporate all or part of the feedback provided by the reviewers. I believe that this work, once achieved a higher level of maturity, can clearly have a place in a top venue like ICLR.

---

### Official Review · Reviewer_Xmnm · 2021-11-03

**Correctness:** 3
**Technical Novelty And Significance:** 3
**Empirical Novelty And Significance:** 2
**Recommendation:** 8
**Confidence:** 4

**Main Review:**

Strengths
- Suitable and persuasive application of the graph and community structure to explain  the activation behavior and relations of the neurons.
- Appropriate usage of the entropy and modularity score to effectively measure the tendency and the predictability of the neuron activation.
- Intuitive hypotheses to validate the correlation between suggested methods and the behavior of the neurons.
Weaknesses
- As authors mentioned in the discussion section(8: Limitations and Future Work). The scalability of this work is suspicious. There ‘s no guarantee that proposed activation analysis methods will also show the same correlation tendency in other neural networks with more complicated architectures than  multi-layer perceptron networks.
- To compute the entropy, activation pattern matrix is required for each layer. The spatial complexity of the matrix is the multiplication of the total number of neurons and the size of a training data. Computational overload will be inevitable when it’s trained on larger data and models.
- Aside from the architecture, the authors only had done experiments on the dropout conditions. Experiments on other various hyperparameters/conditions that can affect neuron configurations or learning ability of the neural network(e.g., pruning) could add more validities to their ‘comprehensive experimental study’.


**Summary Of The Paper:**

This paper examined novel neuron activation pattern analysis on neural network classification models via graph theoretic and entropy-based methods. The authors showed reliability of their work by approving hypotheses on examining the qualitative correlation between model performance and activation patterns. The main technical contribution of this paper comes from explaining the neural classifiers by combining the graph-theoretic and information-theoretic approaches.

**Summary Of The Review:**

The suggested explanatory methodologies and approaches were pretty persuasive to explain the network behavior in microscopic context. Nevertheless, the scalability and practicality of the work should be examined carefully.

---

> ### Author Response · Authors · 2021-11-16
> **Response to Reviewer Xmnm's comments**
>
> We sincerely thank the reviewer for the insightful feedback and for appreciating our approach to explaining deep learning models using graph-theoretic and information-theoretic approaches.
>
> We thank the reviewer for suggesting potential future research to see how the relationship between learning and graph/information metrics varies on models with more complicated architectures than multi-layer perceptron networks, with dropout conditions and other hyperparameters.
>
> In our response to the reviewer's concern over computational complexity, we acknowledge that we have not considered this during our experiments, but we believe it is an excellent point, and in the future, we can work on the algorithm to make it work in real-time during the training process.

---

### Official Review · Reviewer_8T7r · 2021-11-08

**Correctness:** 4
**Technical Novelty And Significance:** 1
**Empirical Novelty And Significance:** 2
**Recommendation:** 3
**Confidence:** 4

**Main Review:**

Strengths:
- The metrics and evaluation procedures are clearly explained. Developing new metrics to measure models activations and relating them to model performance an exciting research direction.

Weaknesses:
- The metrics introduced in this paper do not immediately suggest a way to build higher performance models or provide significant insight into what the model's have learned. Just because a representational metric is correlated with a model's performance does not mean that it causes a model's performance improvement. It also does not mean that a representational metric is predictive of the performance of the fully trained model. I would be more convinced that these are important metrics if they could be incorporated as a regularization term in a loss function (Ex. Selectivity considered harmful: evaluating the causal impact of class selectivity in DNNs (Leavitt & Morcos 2020)) or they could be used to predict which models will generalize better (Ex. Fantastic Generalization Measures and Where to Find Them (Jiang et al. 2019))
- I do not think that the entropy metric is entirely novel. Many past studies have demonstrated that individual units in neural networks become specialized during the course of training for detecting specific features or classes. (Ex. Understanding the Role of Individual Units in a Deep Network (Bau et. al. 2020))
- The finding that modularity is correlated with training accuracy, and that the number of well defined neural communities increase with accuracy could only apply to a vanilla SGD model training setup. Model representations are often very flexible and it could be that regularization against modularity results in a model that is still equally high performance, but is much less modular. Such an experiment could demonstrate if the metrics studied in this paper are measuring an important model representation structure necessary for high performance or a representational coincidence of their particular training method.



**Summary Of The Paper:**

This paper proposes new metrics based on the activation patterns of a model. It propose an activation pattern entropy metric and a graph theoretic metric of neuron communities. The authors then evaluated models trained on MNIST, Fashion MNIST, Fashion MNIST mixed, CIFAR-10, and plant village using their new metrics. They found that entropy is negatively correlated with training accuracy, modularity is correlated with training accuracy, and that the number of well defined neural communities increase with accuracy.

**Summary Of The Review:**

Overall, the metrics in this paper are sensible and clearly explained, but further experiments are necessary to demonstrate insights into what models are learning or what directions for performance improvements the metrics can suggest.

---

> ### Author Response · Authors · 2021-11-16
> **Response to Reviewer 8T7r's comments**
>
> We sincerely thank the reviewer for the insightful feedback and for appreciating our approach to relating the model's activation pattern to the model performance.
>
> The suggestion that we could probably use our performance metrics as a regularization term of the loss function is very interesting. Although the paper's goal was to design graph and information theoretic metrics to improve our fundamental understanding of how the neurons form  community structure, an exploration based on whether or how the performance metrics could be used to enhance the loss functions for better training would be an interesting future avenue to explore. The main difference between our work and the work proposed by Selectivity considered harmful: evaluating the causal impact of class selectivity in DNNs (Leavitt & Morcos 2020) is that we proposed new performance metrics whereas Leavitt & Morcos proposed optimizing the loss functions based on the class selectivity, and in Fantastic Generalization Measures and Where to Find Them (Jiang et al. 2019), Jiang et al. studied the effect of generalization and proposed optimization of models. We would like to thank the reviewer for suggesting the papers and we can evaluate the studies in our future work.
>
> Thanks for pointing out prior research on how a unit becomes more specific to a class. We should revise our paper to discuss this. While previous papers did not design any performance metric based on this observation, we were able to take it a step further by designing an entropy-based measure that clearly distinguishes between well-trained and ill-trained networks.
>
> About the question on generalizability: Our experiments are based on a vanilla SGD model, and exploring whether modularity is a fundamental property that must be required for high performance irrespective of model architectures would be very challenging to establish. Even our activation pattern graph for the vanilla SGD model is non-trivial as it carefully minimizes the impact of redundant edges to have the modularity signature revealed by the standard modularity measures.

---

### Author Response · Authors · 2021-11-21
**Summary of revisions of the documents**

We would like to thank the reviewers for their valuable comments and suggestion. We have edited our documents to reflect the suggestions and the following is a summary of the changes,
1. We have edited the supplementary document to replace the name convolution with fully connected layer (Figure 1, supplementary document).
2. We have added a new section in Related Work (Section 2: Entropy and Deep Learning) which gives an overview of existing works that used entropy in deep learning study and shows how it differs from ours.
3. We clarified the purpose of Hypothesis 3 (H3) in Results and Discussion (Section 7.3: Entropy and Accuracy).
4. We have indicated what an iteration represents in our study (Section 6: Dataset and Model Architecture).
5. We acknowledged that modularity and entropy can be used as a regularization in the loss function in Limitations and Futures Work (Section 8).

---

### Decision · Program_Chairs · 2022-01-20

**Decision:**

Reject

**Comment:**

Understanding neural networks once they have been trained is a big open problem for machine learning. This manuscript designed graph theoretic and information theoretic measures aimed at helping us understand community structure and function in trained networks. In particular, they measure community structure (modularity) and entropy for trained networks and related these to the performance of the networks. The manuscript runs experiments with fully connected networks on problems such as MNIST and CIFAR. Both community structure and entropy measures are shown to correlate (Spearman and Pearson correlation coefficients) with performance metrics in the networks studied.
Reviewers tended to agree that the paper was well written and motivated by an interesting and timely question (understanding trained networks). However, on the whole, most of the reviewers believe that the manuscript is too preliminary for publication at ICLR and I agree. A central issue cited by most of the reviewers is that the experiments are performed on small/toy models for small tasks and under particular hyperparameter regimes. It is therefore unclear to what extent the results would generalize to other situations. E.g. would the results hold for larger dataset or for convolutional neural networks? Connected to this complaint, reviewers worry that there is not enough connection to the literature and baseline methods that could be used to predict performance given measures of trained network activity. Even allowing that the observed correlations are true and generalizable, are these measures better than those covered elsewhere in the literature? Additionally problematic, the measures are not theoretically justified either. Thus, we are missing both reasoned arguments for the metrics and robust quantification beyond a limitted experimental setting. One reviewer, Xmnm, is compelled by the work and recommends acceptance. However, they do not present a compelling case for acceptance, and even repeat several of the concerns raised by other reviewers.
In sum, the work is on an interesting subject and timely, but needs further work to be ready for publication.